# The effect of pregnancy and the duration of postpartum convalescence on the physical fitness of healthy women: A cohort study of active duty servicewomen receiving 6 weeks versus 12 weeks convalescence

David W. DeGroot[1☯], Collin A. Sitler[2☯], Michael B. Lustik[3☯], Kelly L. Langan[4‡], Keith G. Hauret[5‡], Michael H. Gotschall[6‡], Alan P. Gehrich[7☯]*

1 Fort Benning Heat Center, Martin Army Community Hospital, Fort Benning, Georgia, United States of America, 2 Department of Obstetrics and Gynecology, Walter Reed National Military Medical Center, Bethesda, Maryland, United States of America, 3 Department of Clinical Investigation, Tripler Army Medical Center, Honolulu, Hawaii, United States of America, 4 Department of Obstetrics and Gynecology, Madigan Army Medical Center, Joint Base Lewis-McChord, Washington, United States of America, 5 Army Public Health Center, Aberdeen Proving Ground, Maryland, United States of America, 6 Department of Pediatrics, Tripler Army Medical Center, Honolulu, Hawaii, United States of America, 7 Department of Obstetrics and Gynecology, Tripler Army Medical Center, Honolulu, Hawaii, United States of America

☯ These authors contributed equally to this work.
‡ These authors also contributed equally to this work.
* apgurogyn@gmail.com

## Abstract

### Introduction

Pregnancy profoundly affects cardiovascular and musculoskeletal performance requiring up to 12 months for recovery in healthy individuals.

### Objective

To assess the effects of extending postpartum convalescence from 6 to 12 weeks on the physical fitness of Active Duty (AD) soldiers as measured by the Army Physical Fitness Test (APFT) and Body Mass Index (BMI).

### Methods

We conducted a retrospective study of AD soldiers who delivered their singleton pregnancy of ≥ 32weeks gestation at a tertiary medical center. Pre- and post-pregnancy APFT results as well as demographic, pregnancy, and postpartum data were collected. Changes in APFT raw scores, body composition measures, and failure rates across the 6-week and 12-week convalescent cohorts were assessed. Multivariable regressions were utilized to associate risk factors with failure.

**Data Availability Statement:** All relevant data are within the manuscript and its Supporting information Files.

**Funding:** The author(s) received no funding for this work.

**Competing interests:** No authors have competing interests.

## Results

Four hundred sixty women met inclusion criteria; N = 358 in the 6 week cohort and N = 102 in the 12 week cohort. Demographic variables were similar between the cohorts. APFT failure rates across pregnancy increased more than 3-fold in both groups, but no significant differences were found between groups in the decrement of performance or weight gain. With the combined cohort, multivariable regression analysis showed failure on the postpartum APFT to be independently associated with failure on the pre-pregnancy APFT (OR = 16.92, 95% CI 4.96–57.77), failure on pre-pregnancy BMI (OR = 8.44, 95% CI 2.23–31.92), elevated BMI at 6–8 weeks postpartum (OR = 4.02, 95% CI 1.42–11.35) and not breastfeeding at 2 months (OR = 3.23, 95% CI 1.48–7.02). Within 36 months of delivery date, 75% of women had achieved pre-pregnancy levels of fitness.

## Conclusion

An additional 6 weeks of convalescence did not adversely affect physical performance or BMI measures in AD Army women following pregnancy. Modifiable factors such as pre- and post-pregnancy conditioning and weight, weight gain in pregnancy and always breastfeeding were found to be significant in recovery of physical fitness postpartum.

## Introduction

Pregnancy has a profound impact on physical fitness. The physiologic adaptation of pregnancy on the cardiac, pulmonary, renal and musculoskeletal systems has been extensively studied and defined. These changes, in addition to the gestational weight gain and concern for fetal well-being, limit the amount and intensity of exercise that women undertake during pregnancy [1]. Delivery does not allow for rapid return to peak performance levels, as physiologic changes can persist for greater than 1 year after delivery [2]. Even healthy individuals with uncomplicated pregnancies can experience prolonged decrements in physical fitness as compared to their non-pregnant counterparts [3]. The direct impact of pregnancy on physical conditioning is challenging to measure due not only to the difficulty identifying a cohort of women with adequate physical fitness data both before and after pregnancy, but also to the multiple factors beyond the physical that play a role in a woman's recovery from pregnancy. Survey studies and focus groups have identified multiple barriers to postpartum exercise, including limited time, lack of childcare, financial cost of gym memberships and equipment, fatigue, negative self-image, and poor social support [4–6]. These emotional, psychosocial and economic factors are more nuanced but likely as impactful as the physical changes of pregnancy. Full time Active Duty (AD) women who become pregnant while serving in the military provide a cohort in whom some of these challenges to study design are overcome through incentives, established physical fitness standards, regular standardized fitness testing and paid maternity leave. Data from this cohort may provide a more accurate assessment of the impact of pregnancy and delivery on physical fitness, and the length of time it takes for women to recover their fitness postpartum.

The US Army requires all soldiers to maintain physical fitness. Every six months the fitness level of an individual soldier is measured utilizing the Army Physical Fitness Test (APFT); in conjunction with an assessment of body composition. The APFT consists of three events: the push-up, the sit-up and the 2 mile run [7]. These events test the muscular and aerobic

endurance of the soldier. Body composition is screened by measuring the soldiers' height and weight [8]. The soldier who has failed an APFT or does not meet body composition standards is placed into a training program to improve her physical conditioning [9]. While in this status, soldiers cannot attend schooling or be promoted. Recurrent failures lead to separation from the military.

A high percentage of women become pregnant during their time on active duty. Data from the Armed Forces Health Surveillance Branch shows that approximately 13% of service women of child-bearing potential have a pregnancy-related event each year which, in the US Army, equates to 12,000 service women [10]. All AD women are given 6 months following delivery to prepare to undergo an APFT for record in conjunction with body composition assessment [7]. To prepare for this assessment, pregnant AD women are given dedicated time to engage in modified fitness training during pregnancy. After completing their postpartum convalescence and receiving medical clearance, they return to organized fitness training with their units. In February 2016, the US Army implemented a postpartum convalescent leave policy which extended postpartum leave from 6 to 12 weeks [11]. Previously, all AD soldiers had returned to formal physical fitness training, pending medical clearance, 6 weeks after delivery. Following implementation of the revised policy, all women were granted 12 weeks of postpartum convalescence. Regardless of the duration of leave afforded them, both cohorts of women were required to be prepared for the APFT six months after delivery. The question arose whether the duration of postpartum convalescence impacts soldier readiness.

The purpose of this study was to examine the effects of the extended convalescent leave policy on the service woman's physical fitness and body composition. We hypothesized that women in the 12 week convalescent leave group would have lower APFT scores and worse body composition measures postpartum compared to women in the 6 week convalescent leave group. Secondarily, we sought to evaluate the effects of pregnancy on the performance of individual APFT events. Additionally we sought to elucidate risk factors before, during and after pregnancy, that may limit a woman's ability to regain her physical fitness. Lastly, we sought to determine the time necessary for an AD soldier to regain her pre-pregnancy level of physical fitness after delivery.

## Methods

### Army physical fitness test

As described in the Introduction, the APFT consists of three individual events. The push-up event assesses the muscular endurance of the chest and upper arm musculature. Rest during the event is only permitted with arms completely extended. Resting on the ground with any part of the body other than hands or feet results in event termination. The sit-up event assesses the muscular endurance of the core and hip flexor muscles. The sit-up is performed with knees bent, hands clasped behind the head and with another soldier holding the feet down. Rest is only permitted when the spine is perpendicular to the ground. The soldier is not permitted to unclasp her hands from behind her head. These first two events primarily measure muscular endurance, and the score is based on the maximum number of reps performed in two minutes [12]. The 2-mile run is a timed event and measures aerobic fitness. Walking is permitted, but discouraged, and the soldier must complete the run without assistance. Research has reported correlation coefficients of 0.70–0.90 comparing the 2-mile run to a treadmill incremental maximal oxygen uptake test [12]. Each of these events is scored according to age and sex-adjusted standards, whereby more repetitions (push-ups, sit-ups) and faster time (2-mile run) results in a higher score [9]. Failure on any one event, e.g., failure to perform the minimum number of push-up or sit-up repetitions or 2-mile run time greater than the maximum allowable time, is

recorded as a failure for the entire APFT. During the APFT, a soldier's BMI is assessed using height and weight measurements. If a soldier fails to meet the screening standard, percent body fat is estimated using circumferential measures of the neck, hip and waist. These results are compared to age and sex-adjusted standards, and maximum allowable weight varies, based on sex, height and age [8].

## Data collection

Data for this retrospective cohort study were obtained from three independent sources. APFT data was retrieved from a centralized repository within the US Army Digital Training Management System. The APFT data, including raw and adjusted scores as well as the soldier's height and weight is entered into the DTMS system by the soldier's unit following an APFT. This record allows for comparison of APFT results across an extended time frame before and after pregnancy. Compliance with APFT and body composition standards was assessed according to US Army standards [8, 13]. Select pre-pregnancy, antepartum, intrapartum and postpartum data were collected from the inpatient and outpatient electronic medical records (EMR). The de-identified APFT and pregnancy data were compiled in a Microsoft Excel spreadsheet for analysis. The protocol was approved by the Regional Health Command–Pacific Institutional Review Board, which waived the requirement to obtain informed consent. The study was conducted in accordance with federal regulations for the protection of human research volunteers.

## Subjects

All subjects were AD Army women giving birth at a single tertiary Army medical center with a Level 3 neonatal intensive care unit. The study included all women who had delivered their first or second singleton pregnancy of $\geq$ 32weeks gestation between 1 January 2011 and 31 March 2017. Women delivering their third or greater child, or who had a medical condition preventing them from completing the APFT were excluded. A roster of eligible women was generated by the medical center clinical bioinformatics office in November 2017. This roster was then used to extract the appropriate APFT and height/weight data from DTMS, in January 2018. The pre-pregnancy APFT must have been completed within 15 months of the new obstetric registration appointment, and the first postpartum APFT within 15 months of the delivery date. Subjects meeting inclusion criteria were dichotomized into 6- and 12-week convalescent leave groups based on the delivery date.

Demographic and descriptive data included age at time of delivery, race, military rank, marital status and pre-pregnancy height, weight and BMI. Individual patient data were extracted from the electronic medical records from January 2018 through March 2019. Individual Department of Defense Identification numbers were used to link the data from each source; once the master file was compiled, source files were deleted and personal identifiers in the master file were replaced with unique numerical identifiers to preserve anonymity. Obstetric data included gestation age at delivery, weight gain during pregnancy, antepartum complications as well as mode and complications of delivery. The weight at the obstetric registration appointment at 6–8 weeks gestation was used as the baseline pregnancy weight. Weight gain was evaluated according to the Institute of Medicine (IOM) Guidelines for Weight Gain during pregnancy [14]. As these guidelines are based on the body mass index (BMI), height and weight data were used to calculate BMI, which was used for all subsequent analyses. Breast-feeding rates were assessed using the documentation in the neonatal outpatient EMR. Analysis of APFT data was conducted using raw scores (sit-up and push-up reps, and run time) [13]. Scaled scores were used to determine failure rates on the APFT. The primary outcomes were APFT raw scores and BMI measures across the 6-week and 12-week convalescent leave

cohorts. The secondary outcomes were the failure rates on both the APFT events and BMI measurements.

## Statistics

Standard descriptive statistics were used to characterize the cohorts overall and to compare the two convalescent leave groups. Continuous variables are summarized as mean (standard deviation) or median (interquartile range), and also categorized for presentation. Based on sample sizes for the two leave groups and letting alpha = 0.05, a post hoc power analysis showed that the study had 80% power to detect a 5% difference in mean postpartum APFT total scores.

Chi-square tests and Fisher's exact tests were used to assess associations between categorical variables, and unpaired t-tests and nonparametric Wilcoxon rank sum tests were used to evaluate differences between leave groups for continuous variables. Paired t-tests were used for unadjusted analyses to compare postpartum APFT scores with pre-pregnancy levels and McNemar's tests were used for unadjusted analyses to compare failure rates between pre-pregnancy and postpartum APFTs. Analysis of covariance was used for analyses of postpartum APFT scores adjusted for pre-pregnancy scores, and multivariable logistic regression was used for adjusted analyses of postpartum failure rates. A stepwise approach was used to develop adjusted models, with a $p<0.10$ criterion to enter and $p<0.05$ to remove. Demographic factors, obstetric data, BMI, and weight gain during pregnancy were eligible to be included in the models. Age was included in the final model regardless of significance. The majority of variables had no missing data or 1 or 2 records with missing data. BMI at six to eight weeks postpartum was missing for 19 women, marital status was missing for 34, breast feeding status at two months was missing for 60, and BMI measures at the time of APFT was missing for 173 women pre-pregnancy and 166 women postpartum. Because excluding records with missing data may lead to biased results, we used the SAS MI procedure to conduct Markov Chain Monte Carlo multiple imputation to impute 20 data sets with complete data. Logistic regression analyses were subsequently performed on each of the 20 imputed data sets, and parameter estimates were combined using SAS MIanalyze.

Linear repeated measures mixed effects models were used to evaluate changes in mean APFT scores over time, with time as a fixed effect and patient as a random effect. Kaplan-Meier curves were generated to assess time to attain pre-pregnancy fitness levels, treating data as censored if the soldier failed to achieve pre-pregnancy levels by the time of her last recorded APFT. All analyses were conducted using SAS statistical software version 9.4 (SAS Institute, Cary, NC).

## Results

### Demographics

A total of 2103 AD Army women who met initial inclusion criteria delivered singleton pregnancies at Tripler Army Medical Center between 1 January 2011 and 31 March 2017. We identified 460 who met inclusion criteria and had recorded pre pregnancy and postpartum APFT data available for analysis (Fig 1). Eighty percent of the subjects delivered before the new postpartum leave policy had been implemented. Distributions of age, rank and race were similar between the two leave policy groups (Table 1). Ages ranged from 19 to 42 years, with a mean of 25.8 years (4.7) for the 6-week group and 25.8 years (4.2) for the 12-week group. Approximately 3 of 4 women were enlisted soldiers. Unmarried soldiers comprised 22% of the cohort. Approximately four-fifths of deliveries were primiparous, and 22% of deliveries were by cesarean. The BMI distribution was slightly different among the two cohorts. The elapsed time between delivery and the first postpartum APFT was 25 days shorter in the 12-week

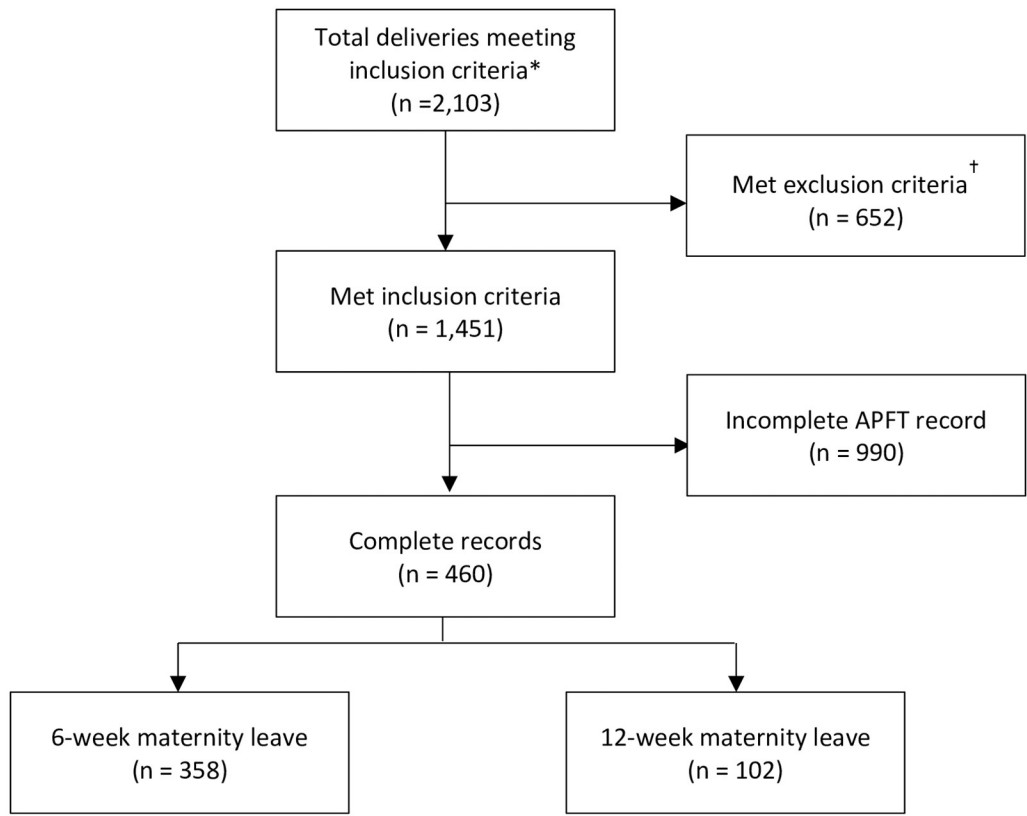

**Fig 1. Selection of study participants.** APFT, Army Physical Fitness Test. * Inclusion criteria are U.S. Army Servicewomen; delivery of pregnancy at Tripler Army Medical Center; delivery between January 1, 2011 and March 31, 2017; and gestational age at delivery greater than or equal to 32 completed weeks. # Exclusion criteria are duplicate record; multiple gestation; parity of three or more. ## Incomplete APFT record included last pre-pregnancy APFT more than 15 months before new obstetric appointment or record of 1st postpartum APFT more than 15 months after delivery in the Digital Training Management System.

convalescent cohort. This cohort also had higher breastfeeding rates at 2 month postpartum. The remaining demographic, social, and pregnancy variables were not different between cohorts. All data collected for this research is contained in the S1 Table.

## Comparison of physical performances across 6-week and 12-week cohort

Both the 6-week and 12-week leave policy cohorts experienced a significant decrement in performance at the first APFT postpartum (APFTPOST) compared to results on their last APFT prior to delivery (APFTPRE) for each APFT event and in overall performance. The decrement of performance across push-ups, sit-ups and run events was equivalent between the two convalescent leave groups (Table 2). Failure rates for the APFT increased from 3.3% pre-pregnancy to 12.4% postpartum for the 6-week leave policy group and from 3.9% to 13.6% for the 12-week leave policy group, ($p < 0.001$ for both), with no difference in the increase in failure rates between leave groups. As a combined cohort, all events and combined scores document significant decrements between APFTPRE and APFTPOST (Table 3). Analysis of failure rates across combined cohort showed significant increases for sit-ups and the 2-mile run but not for push-ups (Fig 2).

**Table 1. Demographic and pregnancy characteristics.**

| Characteristics | All | 6-week leave policy | 12-week leave policy | p-value |
|---|---|---|---|---|
| | (n = 460) | (n = 358) | (n = 102) | |
| | N (%) | N (%) | N (%) | |
| **Age (years)** | | | | 0.695 |
| 19–24 | 216 (47) | 170 (47) | 46 (45) | |
| 25–30 | 169 (37) | 128 (36) | 41 (40) | |
| 31–42 | 75 (16) | 60 (17) | 15 (15) | |
| mean ± std | 25.8±4.6 | 25.8±4.7 | 25.8±4.2 | 0.949 |
| **Race** | | | | 0.477 |
| White | 176 (38) | 133 (37) | 43 (42) | |
| Black | 120 (26) | 93 (26) | 27 (36) | |
| Asian/Pacific Islander | 57 (12) | 43 (12) | 14 (14) | |
| Other | 107 (23) | 89 (25) | 18 (18) | |
| **Rank** | | | | 0.431 |
| Enlisted | 350 (76) | 269 (75) | 81 (79) | |
| Officer | 110 (24) | 89 (25) | 21 (21) | |
| **Marital Status** | | | | 0.109 |
| Single | 93 (22) | 80 (24) | 13 (15) | |
| Married | 333 (78) | 259 (76) | 74 (85) | |
| **Tobacco use** | | | | 0.308 |
| Never | 388 (84) | 297 (83) | 91 (89) | |
| Former | 65 (14) | 55 (15) | 10 (10) | |
| Current | 7 (2) | 6 (2) | 1 (1) | |
| **Parity** | | | | 0.494 |
| 1 | 363 (79) | 285 (80) | 78 (76) | |
| 2 | 97 (21) | 73 (20) | 24 (24) | |
| **BMI at initial OB visit (kg/m$^2$)** | | | | **0.006** |
| <25 | 271 (59) | 220 (61) | 51 (50) | |
| 25-<30 | 164 (36) | 115 (32) | 49 (48) | |
| 30-<35 | 25 (5) | 23 (6) | 2 (2) | |
| mean ±std | 24.5±3.1 | 24.4±3.2 | 24.9±2.7 | 0.100 |
| **BMI at 6–8 weeks postpartum** | | | | 0.103 |
| <25 | 176 (40) | 147 (42) | 29 (31) | |
| 25-<30 | 200 (45) | 149 (43) | 51 (54) | |
| ≥30 | 65 (15) | 51 (15) | 14 (15) | |
| Mean | 26.2±3.7 | 26.1±3.8 | 26.8±3.4 | 0.103 |
| **Weight gain (lbs.)** | | | | |
| mean ± std | 32.9±13.8 | 32.9±14.2 | 32.8±12.2 | 0.937 |
| **Weight gain per IOM guidelines** | | | | 0.694 |
| Adequate | 150 (33) | 120 (34) | 30 (29) | |
| Excessive | 226 (49) | 174 (49) | 52 (51) | |
| Insufficient | 82 (18) | 62 (17) | 20 (20) | |
| **Mode of delivery** | | | | 0.586 |
| SVB | 339 (74) | 261 (73) | 78 (76) | |
| Cesarean | 102 (22) | 83 (23) | 19 (19) | |
| Operative Vaginal | 19 (4) | 14 (4) | 5 (5) | |
| **Gestational age at delivery (weeks)** | | | | 0.197 |
| 30–36 | 29 (6) | 24 (7) | 5 (5) | |

*(Continued)*

**Table 1.** (Continued)

| Characteristics | All | 6-week leave policy | 12-week leave policy | p-value |
|---|---|---|---|---|
| | (n = 460) | (n = 358) | (n = 102) | |
| | N (%) | N (%) | N (%) | |
| 37–38 | 108 (23) | 76 (21) | 32 (31) | |
| 39–40 | 269 (58) | 215 (60) | 54 (53) | |
| 41 | 54 (12) | 43 (12) | 11 (11) | |
| median (IQR) | 39.6 (38.7–40.3) | 39.7 (38.7–40.4) | 39.4 (38.4–40.1) | 0.135 |
| **Fetal weight (grams)** | | | | 0.911 |
| <2500 | 17 (4) | 13 (4) | 4 (4) | |
| 2500–3999 | 406 (88) | 315 (88) | 91 (89) | |
| 4000+ | 36 (8) | 29 (8) | 7 (7) | |
| mean ± std | 3320±460 | 3320±463 | 3323±451 | 0.952 |
| **Breast feeding at 2 months** | | | | **0.030** |
| Always | 204 (51) | 146 (47) | 58 (63) | |
| Sometimes | 80 (20) | 67 (22) | 13 (14) | |
| Never | 116 (29) | 95 (31) | 21 (23) | |
| **Pregnancy Comorbidities** | | | | |
| Shoulder dystocia | 8 (2) | 8 (2) | 0 (0) | 0.209 |
| Gestational Diabetes | 15 (3) | 14 (4) | 1 (1) | 0.209 |
| Hypertensive Diseases of Pregnancy | 57 (12) | 42 (12) | 15 (15) | 0.400 |
| Blood transfusion | 6 (1) | 4 (1) | 1 (1) | >0.999 |
| EBL $\geq$ 1000 (ml) | 7 (2) | 6 (2) | 0 (0) | 0.346 |
| $\geq$ 3rd degree laceration | 13 (3) | 9 (3) | 4 (4) | 0.497 |
| APGAR 1 minute <6 | 23 (5) | 22 (6) | 1 (1) | 0.037 |
| APGAR 5 minute <6 | 6 (1) | 6 (2) | 0 (0) | 0.346 |
| Any complication | 111 (24) | 89 (25) | 21 (21) | 0.431 |
| **Months from delivery to APFT** | | | | 0.088 |
| 2–5 | 50 (11) | 36 (10) | 14 (14) | |
| 6–8 | 242 (53) | 182 (51) | 60 (59) | |
| 9–12 | 126 (27) | 102 (29) | 24 (24) | |
| 13–15 | 42 (9) | 38 (11) | 4 (4) | |
| median (IQR) | 8.0 (6.9–10.3) | 8.2 (7.0–10.9) | 7.7 (6.7–9.4) | **0.007** |

Data are n (%), mean ± std, or median (IQR).

BMI, Body Mass Index; lbs., pounds; IOM, Institute of Medicine; EBL, estimated blood loss; SVB, spontaneous vaginal birth; IQR, interquartile range.

## Analysis of APFT changes across the combined 6-week and 12-week cohorts

Mean APFT raw scores postpartum for each event and overall composite scores did not differ between leave groups (Table 2). Models adjusted for parity, age, BMI at new OB appointment, and pre-pregnancy APFT score showed no differences between leave groups. As no significant difference in postpartum performance was found between the 6-week and 12-week cohorts, risk factor analysis was performed on the combined data. Unadjusted analyses examining risk factors for failure on the APFT show significant effects of demographic, social, pre-pregnancy conditioning and weight factors, but did not show any pregnancy related factors to be significant. Breastfeeding at 2 months was also significantly associated with fewer failures on the

**Table 2. Comparison in each APFT measure across pregnancy between 6-week and 12-week cohorts.**

| APFT Event | 6-Week Leave Group (n = 358) | | 12-Week Leave Group (n = 102) | |
|---|---|---|---|---|
| | **Change from pre-pregnancy** | **Absolute scores** | **Change from pre-pregnancy** | **Absolute scores** |
| Push-ups (reps) | -4.4 ± 11.0 | 36.3 ± 11.9 | -3.6 ± 8.8 | 35.5 ± 10.9 |
| Sit-ups (reps) | -7.0 ± 11.9 | 62.2 ± 12.8 | -6.9 ± 11.9 | 61.5 ± 13.2 |
| 2-mile run (min) | 0.86 ± 1.63 | 18.0 ± 2.0 | 0.93 ± 1.62 | 18.2 ± 1.7 |
| Overall APFT score* | -17.6 ± 32.9 | 240 ± 40 | -19.5 ± 31.4 | 234 ± 38 |
| Weight (lbs.)[†] | 5.3 ± 12.8 | 147 ± 23.5 | 5.5 ± 10.9 | 148 ± 21.6 |

APFT, Army Physical Fitness Test.

Data are mean ± SD.

No APFT events were significantly different across pregnancy between cohorts (p <.05).

* Raw scores of individual APFT events (sit-ups, push-ups, and 2-mile run) are assigned a point value based on age and sex. The overall APFT score is the total point summation of the individual events, and a passing score is 180 points or greater.

[†] Weight change was not recorded for all subjects; n = 172 and n = 34 for 6-week leave group and 12-week leave group, respectively.

APFT. Time to APFTPOST was dichotomized as <9 months vs. ≥9 months after delivery and was not associated with postpartum performance on the APFT (Table 4). Multivariable logistic regression analysis showed that failure on APFTPOST was strongly associated with failure on APFTPRE, failure on pre-pregnancy BMI measurements, elevated BMI at 6 to 8-weeks post-partum, and never breastfeeding (Table 5).

## Analysis of BMI changes across the combined 6-week and 12-week cohorts

APFT height and weight data with calculated BMI were recorded for 229 women at the time of the pre-pregnancy and postpartum APFT. In the 6-week cohort, the failure rates increased from 4.7% to 11.6% (p = 0.008); in the 12-week cohort, the failure rates increased from 2.9% to 11.8% (p = 0.335). Combined BMI failure rates increased from 4.4% pre-pregnancy to 11.7% post (p<0.001). Increases in failure rates did not differ between the 6-week and 12-week leave groups. Using data from the combined cohort, unadjusted analysis of BMI showed female soldiers who were officers, had passed the APFT and BMI standards pre-pregnancy, had weight gain within IOM standards, and had more rapid weight loss in the first 6–8 weeks postpartum,

**Table 3. Raw APFT score and weight comparison before and after pregnancy for combined cohort.**

| APFT Event | Pre-Pregnancy Results (n = 460) | Post-Pregnancy Results (n = 460) |
|---|---|---|
| Push-ups (reps) | 40.3 ± 11.6 | 36.1 ± 11.7 |
| Sit-ups (reps) | 69.0 ± 11.8 | 62.0 ± 12.9 |
| 2-mile run (min) | 17.1 ± 1.6 | 18.0 ± 1.9 |
| Overall APFT score * | 257 ± 30 | 239 ± 39 |
| Weight (kg) [†] | 141.9 ± 21.4 | 147.2 ± 23.7 |

APFT, Army Physical Fitness Test.

Data are mean ± SD.

All post-pregnancy APFT events were significantly different vs pre-pregnancy (p <.001).

* Raw scores of individual APFT events (sit-ups, push-ups, and 2-mile run) are assigned a point value based on age and sex. The overall APFT score is the total point summation of the individual events, and a passing score is 180 points or greater.

[†] Weight was not recorded for all subjects; n = 206 for women with both pre- and post-pregnancy weight.

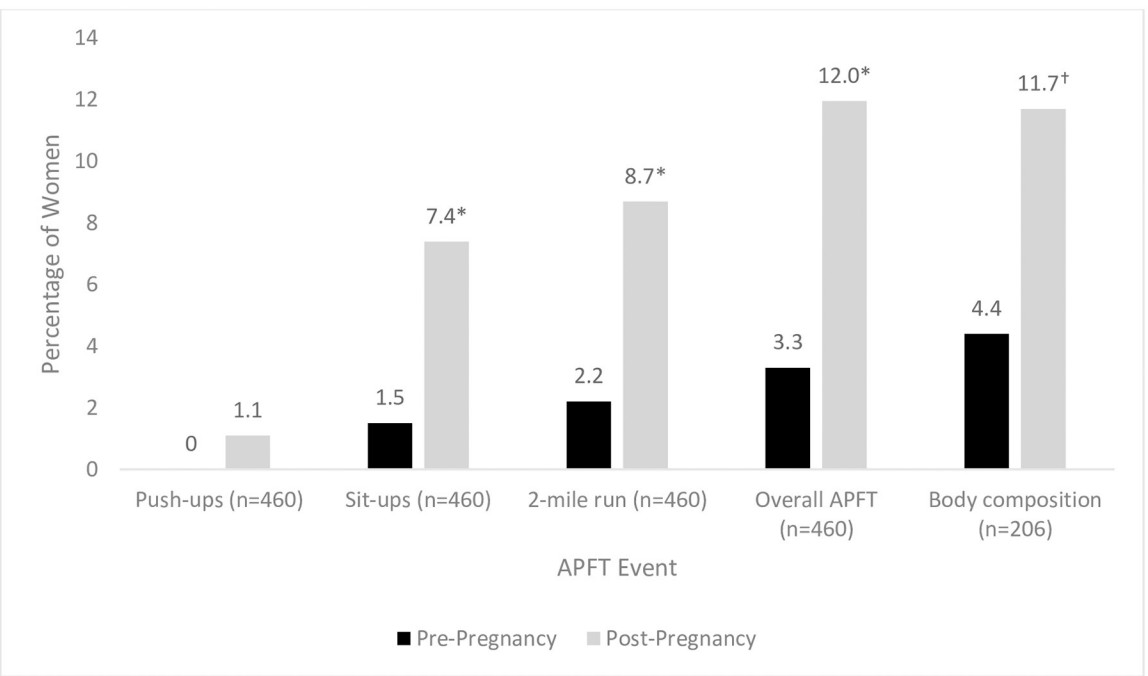

**Fig 2. Army Physical Fitness Test (APFT) and body composition failure rates.** * P<0.01 vs pre-pregnancy.

were significantly more likely to pass BMI standards postpartum (Table 4). Multivariable logistic regression analysis shows age <28 years, BMI >30 kg/m$^2$ at the 6–8 week postpartum appointment and failure of the pre-pregnancy APFT and BMI measurements were significant risk factors for failure on BMI measurements postpartum (Table 5). The mean weight gain comparing AFPTPRE and APFTPOST was 5.3 lbs. with no significant difference between the 6-week and 12-week convalescent leave groups. Mean weight gain between pre- and post-pregnancy APFT was significantly lower for women who were always breastfeeding at 2 months vs. women sometimes or never breastfeeding (2.9 lbs. vs. 7.6 lbs., p = 0.012). Thirty-one percent of all soldiers gained more than 10 lbs. between physical fitness tests across pregnancy. Fifty percent of African American, 21% of Caucasian and 13% of Asian American/Pacific Island soldiers fell into this category. Those soldiers with a BMI ≤25 kg/m$^2$ had an incidence of 33% of excessive weight gain above IOM guidelines whereas soldiers with BMI >25 kg/m$^2$ had an incidence of 70% excessive weight gain above IOM guidelines in pregnancy (p<0.001).

## Duration of pregnancy effect on APFT performance

For the evaluation of the length of time required for an AD soldier to achieve pre-pregnancy physical fitness, the number of postpartum APFTs available for analysis ranged from one to six per individual. The total time interval ranged from 4.5 to 72 months postpartum. Raw scores gradually approached pre-pregnancy levels. On the first APFT postpartum, 25% of women performed equal to or greater than on their sit-up reps and run time, and 37% performed equal to or greater on their push-ups compared to the APFTPRE. Fifty-five percent of women achieved their pre-pregnancy score on their APFT in the 30 months following pregnancy, but the number of push-ups, sit-ups and run times (36.7±12.4 reps, 66.6±12.8 reps and 17.7±1.7 minutes, respectively) remained significantly lower than pre-pregnancy for each event,

**Table 4. Risk factors for failure on APFT and BMI compliance.**

| Risk Factors for Failure on Postpartum APFT | Postpartum APFT | | | Postpartum BMI | | |
|---|---|---|---|---|---|---|
| | PASS *N* (%) | FAIL *N* (%) | P-value | PASS *N* (%) | FAIL *N* (%) | P-value |
| **Social/Demographic Factors** | | | | | | |
| Age (years) | | | **0.036** | | | 0.080 |
| 19–24 | 182 (84) | 34 (16) | | 111 (85) | 19 (15) | |
| 25–30 | 152 (90) | 17 (10) | | 103 (93) | 8 (7) | |
| ≥ 31 | 71 (95) | 4 (5) | | 50 (94) | 3 (6) | |
| Race | | | 0.051 | | | 0.079 |
| White | 160 (91) | 16 (9) | | 109 (94) | 7 (6) | |
| Black | 103 (86) | 17 (14) | | 54 (84) | 10 (16) | |
| Asian/Pacific Islander | 54 (95) | 3 (5) | | 36 (95) | 2 (5) | |
| Other | 88 (82) | 19 (18) | | 65 (86) | 11 (14) | |
| Rank | | | **0.017** | | | **0.005** |
| Officer | 104 (95) | 6 (5) | | 82 (98) | 2 (2) | |
| Enlisted | 301 (86) | 49 (14) | | 182 (87) | 28 (13) | |
| Marital status | | | **0.039** | | | 0.448 |
| Single/divorced | 77 (83) | 16 (17) | | 47 (87) | 7 (13) | |
| Married | 302 (91) | 31 (9) | | 195 (91) | 20 (9) | |
| Tobacco use | | | 0.111 | | | 0.065 |
| Never | 346 (89) | 42 (11) | | 224 (91) | 21 (9) | |
| Former/current | 59 (82) | 13 (18) | | 40 (82) | 9 (18) | |
| **Pre-pregnancy Conditioning** | | | | | | |
| APFT pre-pregnancy | | | **<0.001** | | | 0.053 |
| Passed | 400 (90) | 45 (10) | | 258 (91) | 27 (9) | |
| Failed | 5 (33) | 10 (67) | | 6 (67) | 3 (33) | |
| BMI standards (kg/m$^2$) | | | **0.001** | | | **0.001** |
| Passed | 247 (90) | 28 (10) | | (90) | 19 (10) | |
| Failed | 6 (50) | 6 (50) | | 4 (44) | 5 (56) | |
| **Weight Factors** | | | | | | |
| Pre-pregnancy BMI | | | **0.019** | | | **<0.001** |
| Pre-pregnancy BMI ≥ 25 | 158 (84) | 31 (16) | | 104 (82) | 23 (18) | |
| Pre-pregnancy BMI < 25 | 247 (91) | 24 (9) | | 160 (96) | 7 (4) | |
| Weight gain per IOM guidelines | | | 0.196 | | | **0.004** |
| Within or below guidelines | 209 (90) | 23 (10) | | 136 (95) | 7 (5) | |
| Weight gain above guidelines | 194 (86) | 32 (14) | | 126 (85) | 23 (15) | |
| BMI 6 to 8 weeks postpartum | | | **0.003** | | | **<0.001** |
| BMI ≥ 30 | 52 (80) | 13 (20) | | 32 (68) | 15 (32) | |
| BMI 25 –<30 | 171 (86) | 29 (14) | | 113 (89) | 14 (11) | |
| BMI < 25 | 166 (94) | 10 (6) | | 107 (100) | 0 (0.0) | |
| Weight at 6–8 weeks postpartum compared to 6–8 weeks EGA | | | 0.176 | | | **<0.001** |
| ≥ 15 lbs. | 115 (84) | 22 (16) | | 64 (77) | 19 (23) | |
| 5 –< 15 lbs. | 137 (90) | 15 (10%) | | 97 (98) | 2 (2) | |
| <5 lbs. | 137 (90) | 15 (10%) | | 91 (92) | 8 (8) | |
| **Pregnancy Factors** | | | | | | |
| Pregnancy/Postpartum Comorbidities | | | | | | |
| Shoulder dystocia | 8 (100) | 0 (0.0) | 0.604 | 5 (100) | 0 (0) | >0.999 |
| Gestational diabetes | 13 (87) | 2 (13) | 0.697 | 7 (78) | 2 (22) | 0.231 |
| GHTN/Pre-eclampsia | 47 (82) | 10 (18) | 0.189 | 34 (87) | 5 (13) | 0.570 |

*(Continued)*

**Table 4.** (Continued)

| Risk Factors for Failure on Postpartum APFT | Postpartum APFT | | | Postpartum BMI | | |
|---|---|---|---|---|---|---|
| | PASS N (%) | FAIL N (%) | P-value | PASS N (%) | FAIL N (%) | P-value |
| Blood transfusion | 5 (100) | 0 (0) | >0.999 | 1 (100) | 0 (0) | >0.999 |
| EBL $\geq$ 1000 (ml) | 5 (83) | 1 (17) | 0.537 | 6 (100) | 0 (0) | >0.999 |
| 3rd or 4th degree laceration | 11 (85) | 2 (15) | 0.661 | 8 (100) | 0 (0) | >0.999 |
| APGAR 1 minute < 6 | 20 (87) | 3 (13) | 0.747 | 11 (79) | 3 (21) | 0.161 |
| APGAR 5 minute < 6 | 6 (100) | 0 (0) | >0.999 | 4 (80) | 1 (20) | 0.419 |
| Any complication | 95 (86) | 15 (14) | 0.613 | 60 (87) | 9 (13) | 0.371 |
| Parity | | | 0.289 | | | 0.641 |
| 1 | 316 (87) | 47 (13) | | 205 (89) | 25 (11) | |
| 2 | 89 (92) | 8 (8) | | 59 (92) | 5 (8) | |
| Mode of delivery | | | 0.118 | | | >0.999 |
| SVB | 320 (89) | 38 (11) | | 206 (90) | 24 (10) | |
| Cesarean Birth | 85 (83) | 17 (17) | | 58 (91) | 6 (9) | |
| Newborn weight (grams) | | | 0.374 | | | 0.112 |
| <2500 | 14 (82) | 3 (18) | | 8 (73) | 3 (27) | |
| 2500–4000 | 356 (88) | 50 (12) | | 236 (90) | 25 (10) | |
| >4000 | 34 (94) | 2 (6) | | 20 (95) | 1 (5) | |
| EGA at time of delivery (weeks) | | | 0.452 | | | **0.028** |
| <36 | 10 (77) | 3 (23) | | 6 (67) | 3 (33) | |
| 36–40 | 347 (88) | 46 (12) | | 239 (91) | 23 (9) | |
| $\geq$41 | 48 (89) | 6 (11) | | 19 (83) | 4 (17) | |
| Breast feeding at 2 months | | | **<0.001** | | | 0.503 |
| Always | 189 (93) | 15 (7) | | 119 (92) | 10 (8) | |
| Sometimes | 71 (89) | 9 (11) | | 48 (91) | 5 (9) | |
| Never | 91 (78) | 25 (22) | | 61 (87) | 9 (13) | |
| **Timing of APFT** | | | 0.555 | | | 0.690 |
| <9 months for 1st postpartum APFT | 259 (89) | 33 (11) | | 170 (90) | 18 (10) | |
| $\geq$9 months for 1st postpartum AFPT | 146 (87) | 22 (13) | | 94 (89) | 12 (11) | |

APFT, Army Physical Fitness Test; BMI, Body Mass Index; IOM, Institute of Medicine; EGA, estimated gestational age; GHTN, gestational hypertension; EBL, estimated blood loss; SVB, spontaneous vaginal birth

(p<0.01). By 36 months, 75% of women were able to achieve pre-pregnancy fitness levels (Fig 3a & 3b). Fig 4 shows the Kaplan Meier estimates for time to meet pre-pregnancy APFT levels. The failure rate steadily decreased postpartum from a peak of 13.8% reaching a nadir of 7.7%. The most common individual event failure during the 30+ months was the 2-mile run. Women more quickly regained their push-up scores than those on the sit-up and run events.

## Discussion

The primary findings of this study document that an increase of postpartum leave from 6 to 12 weeks did not adversely affect raw scores on any of the APFT events nor did it decrease the ability of postpartum soldiers to meet physical fitness and body composition standards in the first year postpartum. Risk factors for failure to meet standards on either the APFT or BMI included age <28 years, BMI >30 kg/m$^2$ at the 6–8 week postpartum appointment and failure on the pre-pregnancy APFT and BMI. AD women who successfully met standards postpartum were more likely to meet standards before pregnancy, had gestational weight gain within the

**Table 5. Adjusted odds ratio for risk factors of postpartum APFT and body composition failure with imputed data.**

| Risk Factors for Postpartum APFT failure (n = 460) | Unadjusted Odds Ratio (CI) | P value | Adjusted Odds Ratio (CI) | P value |
|---|---|---|---|---|
| Failed last APFT prior to pregnancy | **9.00 (3.57–22.67)** | **<0.001** | **16.92 (4.96–57.77)** | **<0.001** |
| Failed last BMI measurement (kg/m$^2$) prior to pregnancy | **8.92 (2.64–30.17)** | **<0.001** | **8.44 (2.23–31.92)** | **0.002** |
| BMI at 6 to 8 weeks postpartum * | | | | |
| $\geq$ 30 vs. < 25 | **4.44 (1.86–10.04)** | **<0.001** | **4.02 (1.42–11.35)** | **0.009** |
| 25 –< 30 vs. < 25 | **2.98 (1.41–6.29)** | **0.004** | **2.92 (1.27–6.68)** | **0.011** |
| Breast feeding at 2 months | | | | |
| Never vs. always | **3.49 (1.77–6.88)** | **<0.001** | **3.23 (1.48–7.02)** | **0.003** |
| Sometimes vs. always | 1.54 (0.64–3.70) | 0.339 | 1.71 (0.67–4.35) | 0.261 |
| Age (years) 19–27 vs. 28+[†] | **3.31 (1.46–7.52)** | **0.004** | 2.28 (0.93–5.57) | 0.070 |
| Risk Factors for Postpartum BMI Failure (n = 460) | Unadjusted Odds Ratio (CI) | | Adjusted Odds Ratio (CI) | |
| Failed last APFT prior to pregnancy | **4.77 (1.26–18.13)** | **0.022** | **5.40 (1.19–24.61)** | **0.030** |
| Failed last BMI measurement (kg/m$^2$) prior to pregnancy | **14.24 (3.66–55.39)** | **<0.001** | **6.30 (1.31–30.41)** | **0.022** |
| BMI at 6 to 8 weeks postpartum* | | | | |
| $\geq$30 vs. <30 | **7.86 (3.49–17.68)** | **<0.001** | **8.27 (3.25–21.03)** | **<0.001** |
| Age (years) 19–27 vs. 28+[†] | **3.31 (1.12–9.77)** | **0.031** | **4.82 (1.42–16.41)** | **0.012** |

APFT, Army Physical Fitness Test; BMI, Body Mass Index; CI, confidence interval

Data in bold are statistically significant (p <.05).

*BMI at 6–8 weeks postpartum was dichotomized as <30 vs. 30+ because there were no APFT BMI failures for women whose BMI was <25 at 6–8 weeks postpartum, so odds ratio would be non-estimable for comparisons with the <25 group.

[†]Age was dichotomized as 19–27 vs. > = 28 based on trend seen in unadjusted failure rates vs. expanded age categories (19–21, 22–24, 25–27, 28–30, and >30).

IOM guidelines, and were always breastfeeding at 2 months postpartum. We found great variability among our cohort in the ability of individual soldiers to attain pre-pregnancy fitness standards with some women quickly achieving or even exceeding standards while others never doing so.

The secondary outcomes assessed the effects of pregnancy on the performance of individual APFT events. Women experienced a significant 10% drop in push-up and sit-up repetitions and a 1-minute increase in run time postpartum with a concomitant four-fold increase in the failure rate. These findings are similar to other, smaller studies across military services that reported failure rates between 7–25% on fitness assessments in the first year postpartum [15–19]. We also demonstrated a lower rate of failure on the push-up event postpartum as compared to the sit-up or run, in agreement with Rogers et al. [19]. Based on these results it appears that chest, shoulder and upper arm strength is less likely to be adversely affected by pregnancy than is the core abdominal musculature.

There is limited research documenting how quickly a postpartum woman recovers her strength and cardiovascular conditioning. In a cohort of non-military women, Treuth et al. documented a significant decrement in aerobic and to a lesser degree anaerobic performance at 6 weeks postpartum. At 6 months, strength measures had nearly recovered to pre-pregnancy levels, whereas aerobic assessment measures showed a persistent 10–15% decrement [20]. Studies of trunk flexion showed marked decreases in strength and endurance of isometric contractions postpartum as compared to a nulliparous matched control group at 2 and 6 months postpartum [21, 22]. Our data indicates that core strength decrements persists in the first 9 months postpartum.

As our dataset contained the results of multiple APFT events over an extended period of time, we were able to determine the time course necessary to return to pre-pregnancy fitness

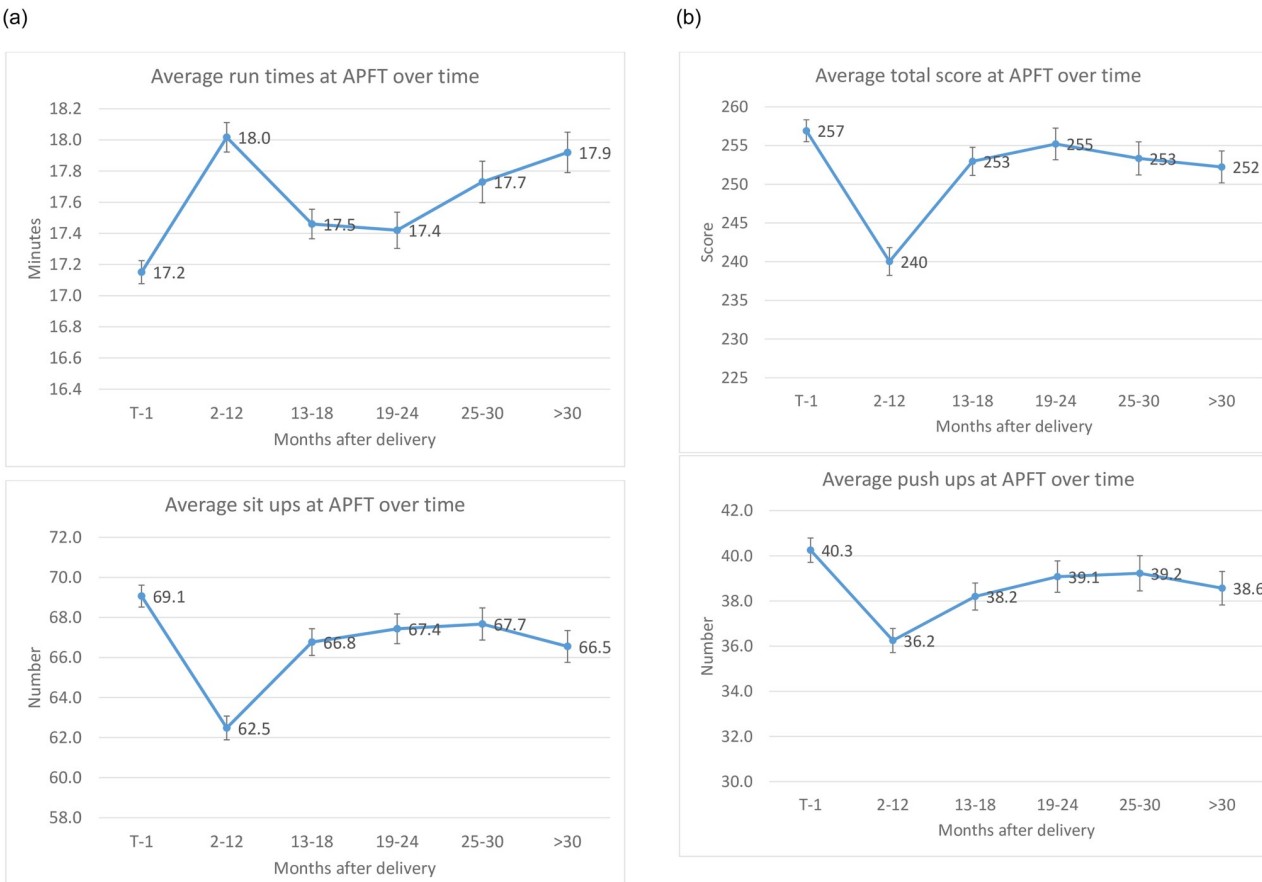

**Fig 3. Postpartum APFT event raw scores and overall scores over time.** APFT, Army Physical Fitness Test. Pre- denotes the last pre-pregnancy APFT. * P<0.01 vs pre-pregnancy.

levels. Rogers et al. analyzed longitudinal data in AD Navy women over a 4-year interval postpartum comparing those to the general population of AD Navy Women. They found that at 2.5–4 years postpartum women were more likely to be overweight, have decreased endurance and decreased core muscle strength as compared to their non -pregnant counterparts [19]. We extend these findings and modeled the time to achieve at least 1 postpartum APFT with a score at least as good as the pre-pregnancy APFT (Fig 4). These data suggest that at 48+ months postpartum 15–20% of AD women will not attain pre-pregnancy levels of physical fitness. Given that some women do rapidly return to pre-pregnancy fitness suggests that lifestyle and/ or motivational factors play a larger role than physiological limits.

Unique to our study is the comprehensive evaluation of potential risk factors which could affect the ability of a mother to recover her physical fitness. Unadjusted risk factors for failure on the postpartum APFT events were pre-pregnancy BMI, postpartum BMI, and failing APFT scores before pregnancy. Based on these findings, the decrement in performance postpartum is not due to specific pregnancy associated conditions or complications, but rather due to a generalized deconditioning affecting strength, anaerobic and aerobic capacity before, during and after pregnancy in healthy active women. These findings indicate that the risk for decreased performance on the APFT postpartum may be modifiable with careful monitoring of weight before and during pregnancy as well as early weight loss following pregnancy.

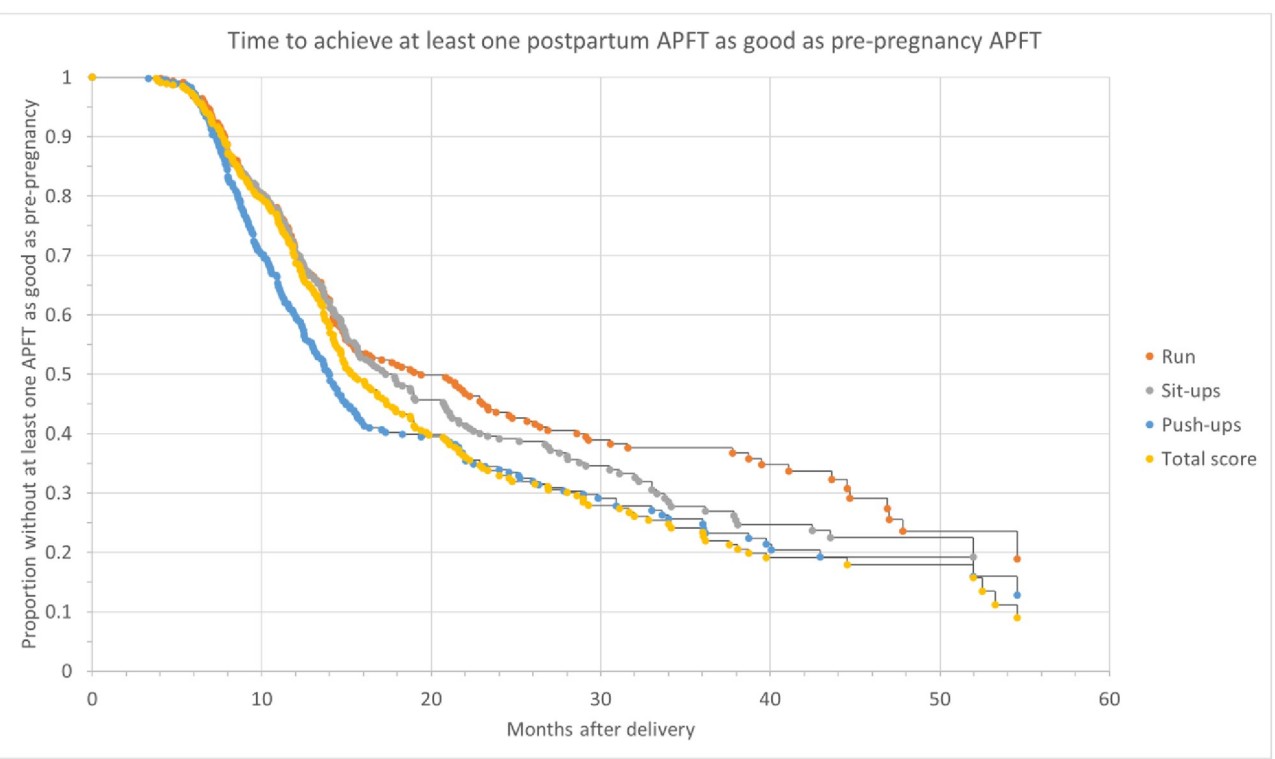

**Fig 4. Kaplan-Meier estimates of time to achieve postpartum individual APFT event and overall APFT score equal to or greater than the last pre-pregnancy APFT.** APFT, Army Physical Fitness Test.

Obstetric factors did not adversely affect postpartum physical performance. Other smaller studies have found gestational diabetes, anemia and cesarean section may impact physical performance postpartum, however none of those factors were significant predictors in our analyses [18, 23].

Research has documented significant benefits of breast feeding on the postpartum mother [24]. The effect of breastfeeding on weight loss and exercise tolerance is mixed in studies on military populations. The consensus, however, is that breast feeding does contribute to weight loss postpartum. Our data shows benefits of breast feeding on physical fitness as mothers who were breast feeding at 2 months had higher pass rates than did those that never breast fed. Although no significant reduction in BMI failure rates were seen in the breast-feeding group, significantly more women who were breast feeding achieved pre-pregnancy weight in the first year postpartum. Additionally, our data indicate that the increased convalescence period instituted in February 2016 was associated with increased breast-feeding rates.

Meeting BMI standards, and by inference losing weight postpartum, is as critical for career advancement among military women as is regaining physical fitness. Although related to fitness, weight loss postpartum has been clearly shown to result from diet and not exercise alone [25]. Women in the military must therefore not only be rehabilitating their strength and aerobic conditioning postpartum, but must actively manage their diet to ensure compliance with standards. Our results document pre-pregnancy weight and weight gain above IOM guidelines as risk factors for failure on BMI measurements at the time of the postpartum APFT. Based on their analysis comparing postpartum Navy and Marine women, Greer et al. also conclude that pre-pregnancy weight is the most critical risk factor for failing postpartum BMI standards

[26]. In a large Navy cohort, Chauhan et al. had similar findings but additionally determined cesarean section to be a risk factor for weight retention at 6 months postpartum [27]. Cesarean delivery was not documented as a risk factor in our cohort at eight and a half months postpartum. Our data also indicates that an age <28 years was an independent risk factor for not meeting BMI standards postpartum. We attribute this to the attrition of less fit women in military service during their early career, thereby selecting for women who are better able to achieve fitness standards postpartum as they grow older within military service.

Incomplete weight loss following pregnancy has been documented among civilian populations to an even greater degree than in military cohorts. Fifty percent of US women gain excessive weight during pregnancy [28]. According to Endres et al., 75% of women do not lose their gestational weight within one year postpartum [29]. In the meta-analysis by Nehring et al., excess gestational weight gain leads to significant weight retention of approximately 10 lbs. at 15 years postpartum, and the SPAWN study documents the deleterious long term effects of this weight retention [30, 31]. As compared to her civilian counterparts, the AD woman is better conditioned before pregnancy, has a lower prevalence of BMI >25 kg/m$^2$, and engages in organized physical fitness training before, during and after pregnancy, all of which is incentivized. Nevertheless, one-third of women retained more than 10 lbs. and one out of eight did not meet BMI standards at the time of their first postpartum APFT. Although not as profound, racial as well as socioeconomic differences mirror those found by Endres et al. [29].

Strengths of our study include the large sample size of healthy women with the accompanying diversity of racial, socioeconomic and age groups which make the results widely applicable to a civilian population. Obstetric and non-obstetric data were carefully compiled to evaluate for potential risk factors. The socioeconomic effects of bearing a child while on active duty are cushioned as AD women receive full pay while on maternity leave. The AD cohort is incentivized and allotted dedicated time to maintain fitness and lose gestational weight. This study is limited by the inherent limitations of a retrospective study with incomplete records and selection biases. The study could not assess the exercise habits and diet of individuals during and after pregnancy nor could it determine the amount of sleep women experienced, all of which can have a profound impact on weight and conditioning [32]. No control group was utilized in this study but normative values have been established and the pre-pregnancy characteristics of our cohort are similar to those found in previously published studies [33, 34]. A large number of women were excluded from our study. The majority of these did not have APFT data entered into DTMS. We hypothesize that this is due to random incomplete data entry and not a selection bias. Lastly, although we attempted to incorporate indirect measures of stress on the new mother to include marital and economic status, this cannot replicate a matched randomized prospective study assessing the complex psychological, social and economic effects of motherhood and ultimately how these affect physical fitness.

## Conclusion

Pregnancy poses significant challenges for a woman's physical fitness. This study documents that women who are granted 12 weeks as opposed to 6 weeks of convalescent leave had no adverse outcome on physical fitness at 9 months postpartum. This study also illustrates the decrement in both aerobic and anaerobic fitness among a healthy and physically active cohort of women and the associated risk factors, some of which are modifiable. Female soldiers who are in better physical condition and of normal weight before pregnancy and at 6–8 weeks postpartum, have gestational weight gain within the IOM guidelines and are breast feeding, are at significantly lower risk of failing the APFT and BMI measures postpartum. These findings can be applied to healthy women desiring to regain their physical fitness after pregnancy.

## Supporting information

**S1 Table. Pregnancy active duty and physical training dataset.**
(XLSX)

## Acknowledgments

Ashley Maranich MD (Pediatrician) for her assistance in developing the method of appropriately assessing breast feeding rates; Ms. Elisha Capstick (Research Assistant) for her assistance in collecting data from the inpatient and outpatient electronic medical records. Ms. Diane Kunichika (Medical Librarian) for her assistance in background research and formatting the manuscript.

## Author Contributions

**Conceptualization:** David W. DeGroot, Kelly L. Langan, Alan P. Gehrich.

**Data curation:** David W. DeGroot, Collin A. Sitler, Michael B. Lustik, Kelly L. Langan, Keith G. Hauret, Michael H. Gotschall, Alan P. Gehrich.

**Formal analysis:** David W. DeGroot, Collin A. Sitler, Michael B. Lustik, Alan P. Gehrich.

**Investigation:** David W. DeGroot, Collin A. Sitler, Michael B. Lustik, Kelly L. Langan, Keith G. Hauret, Michael H. Gotschall, Alan P. Gehrich.

**Methodology:** David W. DeGroot, Collin A. Sitler, Keith G. Hauret, Alan P. Gehrich.

**Project administration:** David W. DeGroot, Alan P. Gehrich.

**Supervision:** David W. DeGroot, Alan P. Gehrich.

**Validation:** Michael B. Lustik.

**Visualization:** Michael B. Lustik.

**Writing – original draft:** David W. DeGroot, Alan P. Gehrich.

**Writing – review & editing:** David W. DeGroot, Collin A. Sitler, Michael B. Lustik, Kelly L. Langan, Keith G. Hauret, Alan P. Gehrich.

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
