## [Decision Letter · Decision Letter 0]

19 Apr 2021

PONE-D-21-04875

The effect of pregnancy and the duration of postpartum convalescence on the physical fitness of healthy women: A cohort study of active duty servicewomen receiving 6 weeks vs 12 weeks convalescence

PLOS ONE

Dear Dr. Gehrich,

Thank you for submitting your manuscript to PLOS ONE. After careful consideration, we feel that it has merit but does not fully meet PLOS ONE’s publication criteria as it currently stands. Therefore, we invite you to submit a revised version of the manuscript that addresses the points raised during the review process.

We look forward to receiving your revised manuscript.

Kind regards,

Antonio Simone Laganà, M.D., Ph.D.

Academic Editor

PLOS ONE

Journal Requirements:

2. In the ethics statement in the manuscript and in the online submission form, please provide additional information about the patient records used in your retrospective study, including: a) whether all data were fully anonymized before you accessed them; b) the date range (month and year) during which patients' medical records were accessed; c) the date range (month and year) during which patients whose medical records were selected for this study sought treatment. If the ethics committee waived the need for informed consent, or patients provided informed written consent to have data from their medical records used in research, please include this information.

Additional Editor Comments (if provided):

The topic of the manuscript is interesting. Nevertheless, the reviewers raised several concerns: considering this point, I invite authors to perform the required major revisions.

Reviewers' comments:

Reviewer's Responses to Questions

**Comments to the Author**

1. Is the manuscript technically sound, and do the data support the conclusions?

Reviewer #1: Yes

Reviewer #2: Partly

Reviewer #3: Yes

2. Has the statistical analysis been performed appropriately and rigorously? 

Reviewer #1: Yes

Reviewer #2: Yes

Reviewer #3: Yes

3. Have the authors made all data underlying the findings in their manuscript fully available?

Reviewer #1: Yes

Reviewer #2: Yes

Reviewer #3: Yes

4. Is the manuscript presented in an intelligible fashion and written in standard English?

Reviewer #1: Yes

Reviewer #2: Yes

Reviewer #3: Yes

5. Review Comments to the Author

Reviewer #1: This study evaluates convalescent period length after delivery on fitness. This was a very well written manuscript.

Abstract: please define AD. Lone p-value are unnecessary.

The introduction was very well written.

Excluding data on covariable is a less robust methods and can lead to biased results. Recommend an alternative method so as to not drop all these participants.

I’m not sure raw scores in the tables are necessary; these could be supplement

Although the findings are well described and if the discussion was just about 6 vs 12 weeks convalescence, this would be great. However, the discussion moved into other ideas as to why the failure rate was at the reported level. Virtually no attention was given to potentially other issues that a woman faces beyond exercise and pre-pregnancy fitness /weight levels. Either reduce the speculation in the discussion or add pieces that were not collected such as the brief mention of sleep, smoking status, etc.

Reviewer #2: I have some questions regarding the methodology and results, as well as some suggestions for clarification of your report.

Abstract

Objective – Here you say your outcome is physical fitness but your purpose statement on Line 98 includes body composition.

Methods – You do not clearly describe here or in the methods section of your paper how body composition was assessed.

Introduction

Lines 51-52 – Can you provide support for your statement here? You seem to be reporting cause and effect, but have no evidence to support it.

Line 54 – Your use of textbooks limits readers’ access to your references. Please replace these with research studies demonstrate the prolonged effect of the physiologic changes you’re describing.

Lines 58-60 – Please provide evidence of the multiple factors (emotional, psychosocial, and economic) you’re describing.

Line 63 – You report that the physical fitness standards are validated. Please cite the validation studies.

APFT – Please provide a description of the individual tests either here or as part of your methods. Describe the procedure and the scoring.

Push-ups and sit-ups – Please provide evidence that they are primarily measures of muscular strength and endurance. Since strength and endurance are not the same, please explain which measures strength and which measures endurance.

2-mile run – Please provide evidence that this is a measure of aerobic fitness. And describe the procedure and scoring.

Line 73 – How is “failure” for each event defined?

Line 74 – You seem to be describing calculation of BMI (height and weight) as body composition measurement. They are not interchangeable. Please provide evidence that circumferential measurements are a valid measure of body fat.

Reviewer #3: This manuscript was to assess the differences in physical fitness of active-duty soldiers who had postpartum convalescence of 6 compared to 12 weeks. The rational is well justified, and results mostly support the discussion and conclusion. I have a few comments.

Introduction: It might not be correct to say "... performance... takes 6 to 12 months to completely normalize". The fact is a good proportion of women take much longer to be relatively back to normal than 12 months.

Under objective, AD needs to spell out.

Why was delivering third or greater child exclusionary?

Results: The elapsed time between delivery and first postpartum APFT was 25 days shorter in the 12-week convalescent cohort. Was this considered in any of the risk factor analysis? This may not seem long, but during postpartum it might make a difference.

Some descriptions of table and figure do not match what are included in the respective table or figure. For example, page 13, line 211; page 14, line 231, page 19, line 296.

On page 18, line 271-274, the description of table 4 does not include BMI at 6-8 weeks postpartum or weight at 6-8 weeks postpartum. However, both were significant.

In table 5, some significant findings are bolded, some are not. Please be consistent.

Tables 4 and 5 include different age categories. Table 4 has 19-24, 25-30, and 31, while table 5 has 19-27 and 28+. No reason was provided for the different classification. Similarly, in table 4, BMI at 6-8 weeks postpartum has 30+, 25-30, and <25, while in table 5, only <30 vs. 30+. Please indicate why.

Throughout the manuscript, units should be added where appropriate including tables and text. For example, years should be added for age...

The authors concluded that pregnancy comorbidities did not influence the comparison results between the 6 and 12 week cohorts. It is noted that these comorbidities were examined individually. However, the small number of cases for each comorbidity might not allow this conclusion to be made. Have the authors considered having any one comorbidity vs. no comorbidity in analyses?

The manuscript may be strengthened by additional analyses examining what factors are associated with time it takes for fitness to return to pre-pregnancy level. Data seem to be available.

6. PLOS authors have the option to publish the peer review history of their article (what does this mean?). If published, this will include your full peer review and any attached files.

Reviewer #1: No

Reviewer #2: No

Reviewer #3: No

---

## [Author Response · Author response to Decision Letter 0]

15 Jun 2021

Alan Paul Gehrich MD COL, MC

Chief, Department of Obstetrics and Gynecology Chief, FPMRS

Tripler Army Medical Center

Antonio Simone Laganà M.D., Ph.D. Academic Editor

PLOS ONE

Dr. Laganà,

Thank-you for giving my team the opportunity to submit revisions of our manuscript to PLOS ONE. In this rebuttal letter, we will respond to the critiques posited by the review of our manuscript entitled: The effect of pregnancy and the duration of postpartum convalescence on the physical fitness of healthy women: A cohort study of active duty servicewomen receiving 6 weeks vs 12 weeks convalescence.

All references to line numbers in the itemized responses refer to the manuscript without track changes.

Journal Requirements:

1. Please ensure that your manuscript meets PLOS ONE's style requirements, including those for file naming. The PLOS ONE style templates can be found at https://journals.plos.org/plosone/s/file?id=wjVg/PLOSOne_formatting_sample_main_body.pdf

We have made the required changes which are documented with track changes.

2. In the ethics statement in the manuscript and in the online submission form, please provide additional information about the patient records used in your retrospective study, including: a) whether all data were fully anonymized before you accessed them; b) the date range (month and year) during which patients' medical records were accessed; c) the date range (month and year) during which patients whose medical records were selected for this study sought treatment

Response: The following has been added to the method section of the manuscript:

“A roster of eligible women was generated by the medical center clinical bioinformatics office in November 2017. This roster was then used to extract the appropriate APFT and height/weight data from DTMS, in January 2018.” (Line 156-158)

“Individual patient data were extracted from the electronic medical records from January 2018 through March 2019. Individual Department of Defense Identification numbers were used to link the data from each source; once the master file was compiled, source files were deleted and personal identifiers in the master file were replaced with unique numerical identifiers to preserve anonymity.” (Line 165-168)

3. If the ethics committee waived the need for informed consent, or patients provided informed written consent to have data from their medical records used in research, please include this information.

Response: A statement clarifying that the IRB waived the requirement for informed consent has been added. (Line 147-148)

Reviewer #1:

1.

2. 3.

Abstract- Please define AD.

Response: AD is the abbreviation for Active Duty Soldiers. These are soldiers who are serving full time in the US Army. The abbreviation has been defined in the abstract and manuscript.

Lone p-values are unnecessary

Response: The 2 lone p-values have been removed from the abstract.

Methods - Excluding data on covariable is a less robust methods and can lead to biased results. Recommend an alternative method so as to not drop all these participants.

Response: We very much appreciate this comment by Reviewer #1 analysis has augmented the results of our study.

In the paper we state:” Records with incomplete data were excluded from analyses for the variable that was missing. The majority of variables had no missing data or 1 or 2 records with missing data. BMI at six to eight weeks postpartum was missing for 19 women, marital status was missing for 34, breast feeding at two weeks was missing for 60, and APFT body composition was missing for 173 women pre-pregnancy and 166 women postpartum.”

In response to this inquiry, we ran models to impute missing data, which affects primarily breast feeding and BMI, for the analyses summarized in Table 5. The method we used to answer the above inquiry is the Markov chain Monte Carlo method (MCMC). For this analysis, the imputation fills in missing data based on relationships with non- missing variables in the database. After applying MCMC, the same variables as before show up as significant predictors of postpartum APFT failure except that the pre- pregnancy pass/fail for HT/WT also becomes significant.

Manuscript Table 5 (pg. 19) presents odds ratios and 95% confidence limits for two outcomes, postpartum APFT failure and postpartum BMI failure. We created tables using the imputed values. These tables are now included in the manuscript.

To describe the imputation approach used, we also added the following statement to the Methods section of the manuscript: “Because excluding records with missing data may lead to biased results, we used SAS MI to conduct Markov Chain Monte Carlo multiple imputation to impute 20 data sets with complete data. Logistic regression analyses were subsequently performed on each of the 20 imputed data sets, and parameter estimates were combined using SAS MIanalyze.” (Line 201-205)

4. Results – I’m not sure raw scores in the tables are necessary; these could be supplement.

Response: We feel that these raw scores contribute important information to the reader, and we therefore wish to keep these in the manuscript. We defer to the editor.

5. Discussion Comments : Although the findings are well described and if the discussion was just about 6 vs 12 weeks convalescence, this would be great. However, the discussion moved into other ideas as to why the failure rate was at the reported level. Virtually no attention was given to potentially other issues that a woman faces beyond exercise and pre-pregnancy fitness /weight levels. Either reduce the speculation in the discussion or add pieces that were not collected such as the brief mention of sleep, smoking status, etc.

Response: We appreciate Reviewer #1’s comments. We do not feel we have speculated extensively in the discussion. In our comment on weaknesses within the study, we state that sleep, diet and physical activity could not be measured by our study. We do not comment on tobacco use, as it was not found to be statistically significant in our initial analysis.

Reviewer #2

1. Abstract

Objective – Here you say your outcome is physical fitness but your purpose statement on Line 98 includes body composition

Response: We clarified abstract to include BMI in the objective

2. Methods – You do not clearly describe here or in the methods section of your paper how body composition was assessed.

Response: We appreciate this critique and have therefore moved a more comprehensive description of the APFT and body composition assessment into the Methods section and abbreviated this description in the Introduction.

“Army Physical Fitness Test: ( new subheading under Methods)

As described in the introduction, the APFT consists of three individual events. Push- ups and sit-ups are scored by the number of repetitions (reps) performed in 2 minutes. The push-up event assesses the muscular endurance of the chest and upper arm musculature. Rest during the event is only permitted with arms completely extended. Resting on the ground with any part of the body other than hands or feet results in event termination. The sit-up event assesses the muscular endurance of the core and hip flexor muscles. The sit-up is performed with knees bent, hands clasped behind the head and with another soldier holding the feet down. Rest is only permitted when the spine is perpendicular to the ground. The soldier is not permitted to unclasp her hands from behind her head. These first two events primarily measure muscular endurance and the score is based on the maximum number of reps performed in two minutes. [12] Walking is permitted, but discouraged, and the soldier must complete the run without assistance. Previous research has reported correlation coefficients from 0.70-0.90 compared to a treadmill incremental maximal oxygen uptake test. [12] Each of these events is scored according to age and sex- adjusted standards, whereby more reps (push-ups, sit-ups) and faster time (2-mile run) results in a higher score. [9] Failure on any one event, e.g. failure to perform the minimum number of push-up or sit-up reps or 2-mile run time greater than the maximum allowable time, is recorded as a failure for the entire APFT. During the APFT, a soldier’s BMI is assessed using height and weight measurements. If a soldier fails to meet the screening standard, percent body fat is estimated using circumferential measures of the neck, hip and waist. These results are compared to age and sex-adjusted standards, and maximum allowable weight varies, based on sex, height and age. [8]” (Line 115-136)

3. Line 51-52 Can you provide support for your statement here? You seem to be reporting cause and effect, but have no evidence to support it.

Response: We have included a reference for this statement.

4. Line 54 – Your use of textbooks limits readers’ access to your references. Please replace these with research studies demonstrate the prolonged effect of the physiologic changes you’re describing.

Response: We have eliminated the textbook references and replaced them with primary sources documenting this effect. (REF 1-3)

5. Lines 58-60 – Please provide evidence of the multiple factors (emotional, psychosocial, and economic) you’re describing.

Response: We have added multiple references to provide evidence for the above. (REF 4-6)

6. Line 63 – You report that the physical fitness standards are validated. Please cite the

validation studies.

Response: Thank you for this comment. As the standards have not been externally validated, we replaced ‘validated’ with ‘established’. (Line 77)

7. APFT – Please provide a description of the individual tests either here or as part of your methods. Describe the procedure and the scoring.

Response: We appreciate this critique. A paragraph has been added to the Methods section detailing the test events and BMI measurements. See response to Reviewer # 2 Item 2

8. Push-ups and sit-ups – Please provide evidence that they are primarily measures of muscular strength and endurance. Since strength and endurance are not the same, please explain which measures strength and which measures endurance.

Response: Due to the number of repetitions necessary to meet minimum standards, both of these events are primarily tests of muscular endurance and reference to strength has been deleted. An appropriate citation has been added as well. (Line 118)

9. 2-mile run – Please provide evidence that this is a measure of aerobic fitness. And describe the procedure and scoring.

Response: Previous research has reported correlation coefficients of 0.70-0.90 between a 2- mile run and a treadmill incremental VO2max test; we have added an appropriate citation.

(Line 121) The test procedure has been described in the Methods section that has been expanded to describe the APFT.

10. Line 73 – How is “failure” for each event defined?

Response: This has been clarified in the description of the APFT event with references (Line 115-136). Further reference is made in the Legend on Table 3.

11. Line 74 – You seem to be describing calculation of BMI (height and weight) as body composition measurement. They are not interchangeable. Please provide evidence that circumferential measurements are a valid measure of body fat.

Response: We agree with the reviewer regarding BMI vs. calculated BMI. Several equations are available in the literature which can be used to estimate percent body fat from circumference measurements. Unfortunately, Army Regulation 600-9, The Army Body Composition Program, does not provide a citation for the specific equations used in the Army.

Reviewer #3

1. Introduction (Abstract) : It might not be correct to say "... performance... takes 6 to 12 months to completely normalize". The fact is a good proportion of women take much longer to be relatively back to normal than 12 months.

Response: The introductory sentence has been changed. “Pregnancy profoundly affects cardiovascular and musculoskeletal performance requiring up to 12 months for recovery in healthy individuals.”

2. Under objective (Abstract), AD needs to be spelled out.

Response: AD – Active Duty has been spelled out

3. Why was the third delivery exclusionary?

Response: We hypothesized that there is a cumulative effect of multiple pregnancies on physical fitness. By examining pre- and post-pregnancy fitness for the 3rd or greater child, we would be introducing a new covariate. This would likely be a small proportion of the total population we studied, making it difficult to control for.

4. Results: The elapsed time between delivery and first postpartum APFT was 25 days shorter in the 12-week convalescent cohort. Was this considered in any of the risk factor analysis? This may not seem long, but during postpartum it might make a difference.

Response: Yes, time between delivery and postpartum APFT was considered as a risk factor for adjusted analyses. As can be seen in table 4 unadjusted, timing of APFT was not associated with either event failure at postpartum APFT (11% for <9 months vs. 13% for >=9 months, p=0.555) or BMI failure (10% for <9 months vs. 11% for >=9 months, p=0.690). Broken down into more categories, event failure rates were 4.0% (2/50), 12.8% (31/242), 15.1% (19/126), and 7.1% (3/42) for 2-5 months, 6-8 months, 9-12 months, and 13-15 months, respectively. Time between delivery and APFT was examined in adjusted models but was not significant, so not retained in final models.

5. Some descriptions of table and figure do not match what are included in the respective table or figure. For example, page 13, line 211; page 14, line 231, page 19, line 296.

Response: The references to Tables on page 13 and page 14 have been corrected. Thank- you for catching this. The figure reference on page 19 was correct.

6. On page 18, line 271-274, the description of table 4 does not include BMI at 6-8 weeks postpartum or weight at 6-8 weeks postpartum. However, both were significant.

Response: Thank-you for recognizing this oversight. The comment was changed to include reference to 6-8week postpartum weight/BMI.

“Using data from the combined cohort, unadjusted analysis of BMI showed female soldiers who were officers, had passed the APFT and BMI standards pre-pregnancy, had weight gain within IOM standards, and had more rapid weight loss in the first 6-8 weeks postpartum,, were significantly more likely to pass BMI standards postpartum (Table 4).”(Line324-328)

7. In table 5, some significant findings are bolded, some are not. Please be consistent.

Response: All results which were found to be significant have been bolded.

8. Tables 4 and 5 include different age categories. Table 4 has 19-24, 25-30, and 31, while table 5 has 19-27 and 28+. No reason was provided for the different classification. Similarly, in table 4, BMI at 6-8 weeks postpartum has 30+, 25-30, and <25, while in table 5, only <30 vs. 30+. Please indicate why.

Response: Age was broken out into three categories in the demographics table to give a general description of the age distribution. Based on unadjusted failure rates for expanded

age categories (19-21, 22-24, 25-27, 28-30, and >30), age was dichotomized as 19-27 vs. >=28 for the risk factor table because the unadjusted failure rates showed a sharp drop after age 27 (19.3%, 13.3%, 13.3%, 4.7%, and 5.3% for the five categories, respectively.

BMI at 6-8 weeks postpartum was dichotomized as <25, 25-<30, and 30+ for Tables 1, 4 and the analysis of postpartum APFT event failure in Table 5. BMI was dichotomized as <30 vs. 30+ in Table 5 for the analysis of postpartum APFT BMI failure because there were no APFT BMI failures for women whose BMI was <25 at 6-8 weeks postpartum, so odds ratio would be non-estimable for the <25 group.

These responses were placed into the footnote section of TABLE 5 to clarify.

9. Throughout the manuscript, units should be added where appropriate including tables and text. For example, years should be added for age...

Response: Units have been added where appropriate.

10. The authors concluded that pregnancy comorbidities did not influence the comparison results between the 6 and 12 week cohorts. It is noted that these comorbidities were examined individually. However, the small number of cases for each comorbidity might not allow this conclusion to be made. Have the authors considered having any one comorbidity vs. no comorbidity in analyses?

Response: This analysis was included in Table 4 in the original submission and was shown at the bottom of Pregnancy/Postpartum Comorbidities section. Analysis of “Any Complication” showed no difference between Pass/Fail in either the APFT or BMI cohorts.

11. The manuscript may be strengthened by additional analyses examining what factors are associated with time it takes for fitness to return to pre-pregnancy level. Data seems available.

Response: We appreciate this recommendation for our manuscript and did perform adjusted Cox proportional hazard analyses in an attempt to address this question. Results, however, identified the same significant predictors that were shown to be associated with failure at first postpartum APFT (e.g. pre-pregnancy levels, age, gravida/parity, and BMI at 6 to 8 weeks postpartum), and so did not appear to add much new. We also are limited in that we do not have data for time-varying covariates that could affect time to return to pre- pregnancy fitness.

Our Kaplan-Meier analysis was intended to be an additional descriptive statistic simply to show how long it may take to achieve pre-pregnancy levels. We have adequate data to show factors affecting the return to physical fitness in the first year postpartum, however we feel our data is not robust enough to evaluate for factors in the 1-3 year postpartum timeframe, and that overanalyzing this data could detract from the manuscript.

This concludes our responses to the reviewer and editor comments for our manuscript. We very much appreciate the review as these have improved the quality of our presentation. All substantial edits are shown in the track changes copy of the manuscript. Once again we thank the team at PLOS ONE for their support of our research.

V/R

Alan Paul Gehrich MD

---

## [Decision Letter · Decision Letter 1]

13 Jul 2021

The effect of pregnancy and the duration of postpartum convalescence on the physical fitness of healthy women: A cohort study of active duty servicewomen receiving 6 weeks vs 12 weeks convalescence

PONE-D-21-04875R1

Dear Dr. Gehrich,

We’re pleased to inform you that your manuscript has been judged scientifically suitable for publication and will be formally accepted for publication once it meets all outstanding technical requirements.

Kind regards,

Antonio Simone Laganà, M.D., Ph.D.

Academic Editor

PLOS ONE

Additional Editor Comments (optional):

I carefully evaluated the revised version of this manuscript.

Authors have performed the required changes, improving significantly the quality of the paper.

Reviewers' comments:

Reviewer's Responses to Questions

**Comments to the Author**

1. If the authors have adequately addressed your comments raised in a previous round of review and you feel that this manuscript is now acceptable for publication, you may indicate that here to bypass the “Comments to the Author” section, enter your conflict of interest statement in the “Confidential to Editor” section, and submit your "Accept" recommendation.

Reviewer #2: All comments have been addressed

Reviewer #3: All comments have been addressed

2. Is the manuscript technically sound, and do the data support the conclusions?

Reviewer #2: Yes

Reviewer #3: (No Response)

3. Has the statistical analysis been performed appropriately and rigorously? 

Reviewer #2: Yes

Reviewer #3: (No Response)

4. Have the authors made all data underlying the findings in their manuscript fully available?

Reviewer #2: (No Response)

Reviewer #3: (No Response)

5. Is the manuscript presented in an intelligible fashion and written in standard English?

Reviewer #2: Yes

Reviewer #3: (No Response)

6. Review Comments to the Author

Reviewer #2: Thank you for your thoughtful responses to my questions and comments. Your revisions are well thought out and detailed. In particular, I commend your interest in this population and believe your findings will make a valuable contribution to the current state of the evidence.

Reviewer #3: (No Response)

7. PLOS authors have the option to publish the peer review history of their article (what does this mean?). If published, this will include your full peer review and any attached files.

Reviewer #2: No

Reviewer #3: No

---

## [Editor Report · Acceptance letter]

19 Jul 2021

PONE-D-21-04875R1 

The effect of pregnancy and the duration of postpartum convalescence on the physical fitness of healthy women: A cohort study of active duty servicewomen receiving 6 weeks versus 12 weeks convalescence 

Dear Dr. Gehrich:

I'm pleased to inform you that your manuscript has been deemed suitable for publication in PLOS ONE. Congratulations! Your manuscript is now with our production department. 

Kind regards, 

on behalf of

Dr. Antonio Simone Laganà 

Academic Editor

PLOS ONE